# Iridoid Derivatives as Anticancer Agents: An Updated Review from 1970–2022

**DOI:** 10.3390/cancers15030770

**Published:** 2023-01-26

**Authors:** Tanaka Ndongwe, Bwalya A. Witika, Nontobeko P. Mncwangi, Madan S. Poka, Phumzile P. Skosana, Patrick H. Demana, Beverley Summers, Xavier Siwe-Noundou

**Affiliations:** 1Department of Pharmaceutical Sciences, School of Pharmacy, Sefako Makgatho Health Sciences University, P.O. Box 218, Medunsa 0204, South Africa; 2Department of Pharmacy Practice, School of Pharmacy, Sefako Makgatho Health Sciences University, P.O. Box 218, Medunsa 0204, South Africa; 3Department of Clinical Pharmacy, School of Pharmacy, Sefako Makgatho Health Sciences University, P.O. Box 218, Medunsa 0204, South Africa

**Keywords:** iridoid derivatives, chemotherapy, cancer agents, isolation, characterization, structure-activity relationship, mode of actions

## Abstract

**Simple Summary:**

Cancer is one of the leading causes of death around the world. The current treatment strategies, especially chemotherapy, have significant side effects. As a result, researchers are investigating alternative treatment strategies, including the use of natural compounds, such as iridoids. Iridoids are compounds predominantly found in plants and, in some cases, insects that have diverse therapeutic uses. These compounds have been reported to have a promising role in the management of cancer.

**Abstract:**

The rise of cancer cases has coincided with the urgent need for the development of potent chemical entities and/or modification of existing commodities to improve their efficacy. Increasing evidence suggests that cancer remains one of the leading causes of death globally, with colon cancer cases alone likely to rise exponentially by 2030. The exponential rise in cancer prevalence is largely attributable to the growing change toward a sedentary lifestyle and modern diets, which include genetically modified foods. At present, the prominent treatments for cancer are chemotherapy, surgery, and radiation. Despite slowing cancer progression, these treatments are known to have devastating side effects that may deteriorate the health of the patient, thus, have a low risk–benefit ratio. In addition, many cancer drugs have low bioavailability, thereby limiting their therapeutic effects in cancer patients. Moreover, the drastic rise in the resistance of neoplastic cells to chemotherapeutic agents is rendering the use of some drugs ineffective, thereby signaling the need for more anticancer chemical entities. As a result, the use of natural derivatives as anticancer agents is gaining considerable attention. Iridoids have the potential to form conjugates with other anticancer, antidiabetic, antileishmanial, and antimalarial drugs, which synergistically have the potential to increase their effects. Published studies have identified the role of iridoids, which, if fully explored, may result in cheaper and less toxic alternative/adjuvant cancer drugs. The subject of this article is natural and synthetic iridoid derivatives and their potential therapeutic roles as anticancer agents.

## 1. Introduction

Cancer is one of the leading causes of death globally, with a recurrent exponential increase in new cases annually. Out of 183 countries, 112 countries reported that cancer is either the first or the second major cause of death in individuals below the age of 70 [1]. It has further been reported that cancer cases, such as colon cancer, are likely to double by 2030, with a recent report from the World Health Organization (WHO) (2020) quoting 10 million deaths per year [2,3]. The increase in cancer cases is ascribed to genetic predisposition, diets, and lifestyles that make individuals susceptible to cancer [3,4]. Despite the skyrocketing cancer cases, current treatment methods, such as chemotherapy, surgery, and radiation, come with debilitating shortcomings, including devasting side effects [5]. The general concern associated with the use of traditional anticancer drugs is their low bioavailability, which prevents them from penetrating biological membranes [6]. For instance, orally administered anticancer drugs mostly suffer significantly low bioavailability under acidic conditions and low diffusion across gastrointestinal tract epithelial cell membranes [6,7,8]. As such, greater doses are typically required due to their limited absorption in order to achieve the desired efficacy [6]. Chemotherapy and radiation have side effects such as anemia, bleeding, and alopecia that may deteriorate the condition of the patient in some instances [9,10]. Additionally, surgery has a major drawback, as its application is mostly limited to the early stages of cancer [11]. Furthermore, the current cancer treatments are generally expensive, making it extremely difficult for most cancer patients to access the available treatments.

To find alternatives for cancer management, phytochemistry research has recently increased its focus on natural compounds, such as iridoid derivatives and their potential role in numerous diseases, including cancer [12,13]. These compounds are mainly distributed in plants and easily react with sugar to form glycosides, partly due to their unstable nature [14]. Plants from families such as the Apocynaceae, Gentianaceae, Lamiaceae, Loganiaceae, Rubiaceae, Scrophulariaceae, and Verbenaceae [15] and insects, including butterflies, have been reported to contain iridoids [16]. Iridoids are chemical compounds that form part of a type of monoterpenoid in the general form of cyclopentanopyran found in a wide variety of plants and some animals [17].

In most plants, iridoids are primarily found in angiosperm dicotyledons and, in some cases, monocotyledons, where they are present in some leaves, fruits, roots, and sprouts [15,18,19]. It has been reported that iridoids play a protective role against predators in both plants and animals. This is partly due to their bitter taste, which seems to discourage predators [20,21]. To date, iridoids have been identified as key compounds in Chinese natural medicine used in the treatment and management of many diseases, including inflammatory conditions [22]. About eight families of insects have been reported to contain iridoids, and some of the insects include *Anisomotpha buprestoides*, *Phaedon amoraciae*, *Phaedon cochleariae*, *Gastrophysa viridula*, *Plagiodera versicolora,* and ants *Iriabmyrmex detectus* [23].

Iridoids have been extensively researched for their anti-tumor properties and therapeutic benefits in chronic conditions [14,24,25]. Recent reports have shown that ether terpenes can also impede DNA polymerase activity, indicating that new anti-tumor drugs derived from iridoids could be created to prevent DNA replication in cancer cells [26,27,28]. As extraction techniques and storage conditions have improved over the past few years, more and more iridoids have been discovered [29,30,31,32]. A review of studies on iridoids further showed that these compounds could be a major focus of research in a variety of areas, including medicine and the pharmaceutical industry. By enhancing antioxidant defenses and blocking cascades, they specifically target neurotoxicity and oxidative stress, which is essential in disease treatment [29,33,34,35,36]. Furthermore, recent insights into iridoid research show that their key biological activity is ascribed to the inhibition of the expression of multiple important proinflammatory proteins, thereby exhibiting a variety of anti-inflammatory actions [37].

At present, there are prominent examples of iridoids that have attracted significant attention in cancer research, such as aucubin (**1**) and catalpol (**2**), which have been found to have diverse pharmacological profiles [13]. Published studies have reported the anticancer and anti-inflammation properties of aucubin (**1**) and catalpol (**2**), making them promising chemical entities [13,38]. Additionally, most iridoids are of botanical origin and some seem to have an acceptable safety profile, which may necessitate the search for cancer treatments with minimal side effects [13]. Furthermore, the surfacing of synthetic iridoids is also gaining considerable attention in cancer, making the role of iridoids indispensable in cancer treatment, as they are likely to contribute to the discovery of hit or lead compounds for cancer management [39]. In light of the above, there is a need to search for potent chemical entities for the management of cancer. The subject of this article is natural and synthetic iridoid derivatives and their potential therapeutic roles as cancer agents.

## 2. Literature Sources and Search Strategy

To compile all of the relevant information about iridoids, their occurrence, isolation, and structure, in vivo and in vitro studies of anticancer properties, and structure–activity relationships, the following online databases were used: SciFinder, Scopus, Science Direct, Springer Link, PubMed, and Google Scholar. The databases were used to search for articles published relevant to the topic. The keywords used were iridoids anticancer activity OR iridoids antitumor activity OR iridoids anticancer structure–activity relationship. The articles considered were published between the years 1970 and 2022.

## 3. Bibliometric Studies

To date, no bibliometric studies accessible to the authors in this regard have been published. However, most studies appear to be predominantly from Asia [40,41,42,43,44,45,46,47,48]. Seemingly, other countries have published only a few studies, including the United States of America, South Africa, and Brazil [49,50,51]. Chinese herbs have mostly been investigated for their potential role in numerous conditions, as their use dates back centuries. Some of the leading plant families that are rich in iridoids are the Gentianaceae, Valerianaceae, and Cornaceae. Most of the recurrently mentioned plant species arise from the aforementioned plant families and have served as reliable sources of iridoids [14].

## 4. Different Types of Iridoids

Natural iridoids are obtained from plants and animals as specialized metabolites. Structurally, iridoids are cyclopentan-(e)-pyran monoterpenoid compounds [17,52] derived from acetal and have a complex chemical structure. The chemical structure of iridoids is based on the iridane skeleton (1-isopropyl-2,3-dimethylcyclopentane) and the contained cyclopentane rings are, in many cases, fused to six-membered heterocycles, such as dihydropyran rings. The leading types of iridoids commonly recognized are glycosidic iridoids, non-glycosidic, secoiridoids, and bis-secoiridoids iridoids (also called dimer iridoids), and these differ as a result of their different biosynthetic pathways [14,52]. Figure 1 depicts the basic structure of iridoids.

Glycosidic iridoids, such as aucubin (**1**), have a glycosidic bond as the key feature, which is mostly present at aglycone C-1 or C-11 hydroxyl. They can be further subdivided into five groups depending on the number of carbon atoms on the skeletal systems, which range from 8 to 10 and 14, and alkaloid-conjugated iridoid glycosides [15]. The non-glycosidic-bond iridoids are characterized by an iridane skeleton/cyclopentane ring as their distinguishing feature and are made up of iridoid lactones and iridodials, while bis-iridoids are formed by the dimerization of iridoids and secoiridoids [36,37]. Secoiridoids are formed by the cleavage of the cyclopentane ring. The diversity of iridoids is partly due to the differences in their biosynthetic pathways, which incorporate multiple reactions, such as phosphorylation, cyclization, oxidation, and glycosidation [14,53,54].

## 5. Occurrence of Iridoids

Iridoids are found in dicotyledon angiosperms and, in some cases, monocotyledons [15]. The isolation of iridoids from plants began in 1848, though the structural elucidation commenced in the 1950s [55]. Quantitatively, iridoids differ depending on the type of plant and the parts they are present in. They are mostly found in the leaves, fruits, roots, and, in some cases, rhizomes [15,39]. Currently, around 60 plant families have been reported to produce iridoids, with plant orders such as the Dipsacales, Gentianales, and Lamiales having significantly high concentrations of iridoids [15]. Some of the discovered iridoid glycosides include aucubin (**1**), catalpol (**2**), and antirrhinoside (**3**) which are mostly present in the following plant families: Scrophulariaceae, Vebananaceae, Bignoniaceae, Oleaceae, and Plantaginaceae [15,56,57]. The occurrence of iridoids should be carefully considered, as some of the techniques used in extraction may pose a significant threat to the extinction of some species.

## 6. Isolation and Characterization of Iridoids

Natural iridoids are obtained from medicinal plants and animals using different extraction and isolation techniques, while synthetic iridoids are derived from chemical reactions using different reaction pathways [29,58]. Over 600 iridoids have been isolated and described from different plant families [59]. As advanced isolation techniques are being developed and implemented, the number of natural iridoids isolated has recently increased significantly. Some of the widely used techniques in the separation and isolation of iridoids are thin-layer chromatography (TLC), column chromatography, and high-performance liquid chromatography (HPLC) [60,61].

### 6.1. Solvent Systems

The most prominent solvent systems usually used in the isolation of iridoids are methanol, ethanol, butanol, and water [62,63,64]. These solvents are either used alone or in combination, depending on the polarity of the target compound to achieve optimum isolation [63,65]. However, it should be noted that the solvents used may possess health concerns, as some are corrosive, carcinogenic, and may also be toxic when released into the environment.

### 6.2. Thin-Layer Chromatography (TLC)

The TLC technique, though ancient and tedious, has long been used in the isolation of iridoids. The technique is primarily used in purification with solvents and the identification of retention factor values and functional groups present [54,66]. Additionally, TLC is used for the purification and identification of iridoids. However, the TLC technique has limitations, such as partial decomposition and irreversible adsorption, and may not be favorable, especially when considering the total yield of the pure compound [67].

### 6.3. Column Chromatography (CC)

A considerable body of literature has reported the use of the column chromatography technique in the isolation of iridoids [68,69,70]. The technique is used to separate compounds by using two phases, namely the stationary phase and the mobile phase [71]. For instance, Zhou et al. (2007) reported findings regarding an iridoid, 6′-*O*-sinapoylgeniposide (**4**), isolated from *Gardenia jasminoides* used for the prevention or treatment of nervous system degenerative disorders, such as dementia. The reported iridoid was isolated using silica gel and column chromatography techniques. For the elucidation of the structure, techniques comprising 1D and 2D nuclear magnetic resonance (NMR) and high-resolution electrospray ionization–mass spectrometry (HR-ESI-MS) are used [72]. However, the isolation of iridoids using CC is generally time-consuming and not effective; hence, other isolation techniques have been explored to circumvent the shortcomings of CC [67].

### 6.4. High-Performance Liquid Chromatography (HPLC)

HPLC, on the other hand, is more frequently used in the isolation of iridoids, and it is also essential in their purification and quantification [73]. Xu et al. (2012) isolated two unreported iridoids, namely jatadoids A (**5**) and B (**6**), along with known compounds. The iridoids were isolated using techniques such as HPLC analysis and silica gel column chromatography. The chemical structure of the isolated iridoids was elucidated using IR, electrospray ionization–mass spectrometry (ESI-MS), HR-ESI-MS, and 1D and 2D NMR [74]. A phytochemical study by Kim et al. (2009) identified two glycoside iridoids, i.e., 4‴-*O*-β-D-glucopyranosyltrifloroside (**7**) and 4‴-*O*-β-D-glucopyranosylscabraside (**8**), together with three known secoiridoids, including trifloroside (**9**), scabraside (**10**), and gentiopicroside (**11**) from the rhizomes of *Gentiana scabra*. Column chromatography techniques were used in the isolation and purification of some, while HPLC was used for purification [75]. The isolated compounds were then identified using spectroscopic analyses, namely ultraviolet (UV), IR, MS, 1D, and 2D NMR. Jiang et al. (2005) used the HPLC technique to isolate the iridoid gentiotrifloroside together with six known compounds, including loganic acid (**12**), 6-*O*-β-d-glucopyranosylgentiopicroside (**13**), swertiamarin (**14**), gentiopicroside (**11**), and sweroside (**15**) from *Gentiana triflora* and *Gentiana rigescens* [60].

### 6.5. Modern Isolation Techniques

Recently, more advanced and intricate methods have been developed, and these include droplet counter-current chromatography (DCCC), high-speed counter-current chromatography (HSCCC), gas–liquid chromatography (GLC), and capillary zone electrophoresis (CZE) [70,76]. Wu et al. (2009) identified 11 iridoids in which 5 derivatives were unreported, and these were 2,3,6-tri-O-acetyl-40-O-trans-p-(O-b-D-glucopyranosyl)coumaroyl-7-ketologanin (**16**), 2′-O-caffeoylloganic acid (17),2-O-p-hydroxybenzoylloganic acid (**18**), 2-O-trans-p-coumaroylloganic acid (**19**), and 2-O-cis-p-coumaroylloganic acid (**20**) isolated from whole plants of *Gentiana loureirii*. The six known iridoids were 7-ketologanin (**21**), loganic acid (**12**), sweroside (**15**), boonein (**22**), and isoboonein (**23**), which were isolated with three other known compounds. To successfully isolate iridoids, combined chromatography techniques, including silica gel and octadecylsilyl (ODS) CC, were used, while spectroscopic techniques, such as 2D NMR, were used in structural elucidation [77].

The aforementioned isolation techniques use various extraction methods with varying degrees of success in isolating iridoids. As a result, the percentage yields and recovery rates are equally impacted.

Below, some of the extraction techniques are succinctly summarized to provide an understanding of the separation strategies and their respective effectiveness in isolating iridoids from plants. The extraction conducted by Maurya et al. (2020) reported that roots weighing 4.0 kg obtained from *Valeriana jatamansi* yielded 570 g of a semi-solid residue from the extracts. An amount of 520 g of the extract was suspended in deionized water and extracted using different solvents. Seven iridoid derivatives were further isolated from the extract and accounted for a total mass of 568.2 mg [78]. Similarly, Bai et al. (2018) carried out a study where 1.5 kg of plant material was extracted with chloroform and ethyl acetate to yield 20.0 g of crude extract. The purification of two of the four fractions obtained from the first CC resulted in the identification of three compounds, including cymdahoside A (**24**), catalpol (**2**), and ajugol (**25**), weighing 21.0 mg, 18.0 mg, and 24.0 mg respectively [79]. The drastic decline in the extraction yield to the final amounts of pure compounds is a clear indication of the limitations of such isolation techniques.

However, techniques such as the counter-current chromatography method and speed counter-current chromatography appear to be more effective not only at recovering the mass, but also in obtaining pure iridoids. For instance, a study by Wang et al. (2015) reported a fraction weighing 1000 mg, which resulted in the isolation of four iridoids, namely gardenoside (**26**), 6β-hydroxy geniposide (**27**), geniposidic acid (**28**), geniposide (**29**), and crocetin derivatives, including crocin-1, crocin-2, crocin-3, and crocin-4, with the following respective yields in some fractions of 151.1 mg, 52.2 mg, 24.5 mg, 587.2 mg, 246.2 mg, 34., 24.4 mg, and 24.7 mg, with a percentage purity ranging between 91.7% and 98.9% [73]. Another study that used a similar technique on a powdered sample of *Fructus Corni* weighing 1.0 kg yielded 45 g of the pure compound. Subsequently, the yield of 100 mg taken for further fractionation produced 12.6 mg of loganin (**30**), 5.9 mg of sweroside (**15**), and 28.5 mg of morroniside (**31**) (purity of 98.6%, 97.3%, and 99.1% with a total recovery of 90.4%, 91.8%, and 89.1%, respectively) [76]. Similarly, an improvement in the isolation of iridoids, such as geniposide (**29**) catalpol (**2**) catalposide (**32**), and verproside (**33**), was obtained through speed counter-current chromatography [30,80,81].

Researchers are not capitalizing on the advanced techniques used in the isolation and purification of iridoids. Most recent studies still incorporate tedious and less effective techniques, such as CC which still has significant shortcomings, such as lower purity, poor recovery rate, and requiring a lot of time [70]. Researchers may consider developing a system of appraising isolation techniques from the published studies to gain new insights into fractionation, isolation, purification, characterization, CCC, and HSCCC. Additionally, studies should consider techniques that are less time-consuming and require fewer solvents while producing higher yields and achieving sample recovery with the highest percentage purity of compounds.

## 7. In Vitro and In Vivo Anticancer Studies of Iridoids and Structure-Activity Relationships (SAR) Studies

In vitro and in vivo studies have played a pivotal role in broadening our understanding of the pharmacological profile of iridoids. In addition, structure-elucidation techniques have greatly assisted in depicting the functional groups responsible for the pharmacological profile of iridoids [58]. Research studies have covered the cytotoxic, cytostatic, and antiproliferative properties of iridoids against cancer cell lines. Of interest are the iridoids that were found to have biphasic anticancer properties by Saracoglu et al. (2012) [82]. The results of the in vitro study demonstrated that some iridoids from *Veronica* species, namely verminoside (**34**), amphicoside (**35**), and veronicoside (**36**), were cytotoxic against the Hep 2 cell line, with IC_50_ values of 128 µM, 340  µM and 153.3 µM respectively while acetylcatalpol (**37**), aquaticoside B (**38**) and C (**39**), catalposide (**32**), veratroylcatalposide (**40**), and verproside (**33**) had cytostatic effects against Hep 2 cell lines at 100 µg/mL. Furthermore, verminoside (**34**) also displayed notable toxic effects when tested against RD (IC_50_ = 70 µM) and L20B (IC_50_ = 103 µM) cell lines. All of the iridoids investigated were cancer-selective, as they were not cytotoxic in the Vero cell line (normal cells).

A recent in vitro study on an iridoid-rich extract from the genipap fruit (ripe and unripe) of *Genipa americana* conducted by Neri-Numa et al. (2020) evaluated the anticancer properties of extracts containing geniposidic acid (**28**), gardenoside (**26**), genipin-1-β-gentiobioside (**41**), geniposide (**29**), 6″-*O*-*p*-coumaroyl-1-β-gentiobioside geniposidic acid (**42**), 6″-*O*-*p*-coumaroylgenipin-gentiobioside (**43**), genipin (**44**), 6′-*O*-*p*-coumaroyl-geniposidic acid (**45**), and 6′-*O*-feruloyl-geniposidic acid (**46**). The extracts were tested against ion glioma and breast cancer cell lines (glioma, CNS), MCF-7 (breast), NCI-ADR/RES (breast expressing the multiple-drug-resistance phenotype), 786-0 (renal), NCI-H460 (lung, non-small cells), PC-3 (prostate), HT-29 (colon), and K562 (leukemia). Among the two extracts tested for anticancer properties, the unripe genipap fruit, which mostly contained genipin (**44**), was found to have anti-proliferative and anticancer effects against both glioma and breast cancer cell lines, with sample concentrations required to achieve 50% inhibition of cell proliferation GI50 (μg/mL) of 29.6 and 29.7. The genipin (**44**) standard was observed to inhibit all cell lines evaluated [19].

Furthermore, another in vitro study evaluated seven isolated compounds in which four iridoids were previously unreported. The isolated compounds from the roots of *Valeriana dioscoridis* were dioscoridin A (**47**), 1-epi-bosnarol (**48**), 8-epi-deoxyloganin aglycone (**49**), dioscorin B (**50**), dioscorin C (**51**), 10-acetylpatrinoside (**52**), and 10,2′-diacetylpatrinoside (**53**)**.** These compounds were investigated for anticancer activity against three human cancer cell lines of gynecological origin (HeLa, A2780, and T47D) using an MTT assay. Among the isolated compounds, it was reported that 1-epi-bosnarol (**18**) had the highest anticancer activities, while the others showed moderate anticancer activity (at 10, 30, and 60 μM concentration) when compared with cisplatin (Kırmızıbekmez et al., 2018) [83].

An unreported iridoid, prismatomerin (**54**), was isolated by Krohn et al. (2007) together with an already known iridoid glucoside, gaertneroside (**55**). The isolated compounds were then tested for anticancer properties against MCF7 (breast cancer), NCl-H460 (lung cancer), and SF-268 (CNS cancer). When tested for anticancer properties in 60 human tumor cell lines, prismatomerin (**54**) was found to have high inhibition in leukemia cell lines (GI50 < 10 nM) [84].

A study by Apisornopas et al. (2018) on the synthesis of cytotoxic compounds from the natural iridoid glycoside (durantoside I) (**56**) obtained from *Citharexylum spinosum* reported some promising anticancer properties. The synthetic analogues were synthesized by reactions including silylation, acetylation, and decinnamoylation. While 13 durantoside I analogues were synthesized, only two compounds were found to be more cytotoxic when compared with durantoside I (**56**) in all of the cell lines. A structural comparison of the compounds showed that the silyl-tert-butyldiphenylsilyl compound devoid of the cinnamate group was more cytotoxic than the compound with the cinnamate group at carbon 7 [85].

The anticancer efficacy of plumericin (**57**), a natural iridoid isolated from *Momordica charantia*, was evaluated in vitro against leukemic (NB4 and K562), breast cancer (T47D) and liver cancer (C3A), normal human fibroblast (HF), and lung cancer (A549) cell lines. The results obtained demonstrated that plumericin (**57**) displayed considerable antiproliferative activity against two leukemia cell lines with effective doses (ED_50_) of 4.35 ± 0.21 and 5.58 ± 0.35 μg/mL [86].

Another study that sought to identify anticancer compounds isolated three iridoids, i.e., reptoside (**58**), 8-acet-ylharpagide (**59**), and harpagide (**60**), from *Ajuga decumbens*. Among the three isolated compounds, 8-acet-ylharpagide (**59**) had significant anticancer effects on EBV-EA induction and on mouse tumorigenesis. Additionally, 8-acet-ylharpagide (**58**) was evaluated for antitumor effects on mouse hepatic tumors, where 8-acet-ylharpagide (**59**) in the test groups was administered orally before and after treatment and for a period of 20 weeks [87]. From the results, it was established that 8-acet-ylharpagide (**59**) exhibited remarkable inhibitory effects in the two-stage carcinogenesis test.

An in vitro and in vivo study on mice with leukemia cancer by Isiguro et al. (1986) evaluated iridoids (glycosides and aglycones). Iridoid glycosides, namely aucubin (**1**), scandoside methyl ester (**61**), geniposide (**29**), loganin (**30**), sweroside (**15**), gardenoside (**26**), and gentiopicroside (**11**), were investigated for their anticancer properties. All of the iridoids demonstrated no anticancer activity. However, the aglycones of the iridoids, especially aucubin (**1**) and scandoside methyl ester (**61**), had significant anticancer activity against leukemia P388, with maximum total/control values of 162% and 160%, respectively, at 100 mg/kg. Scandoside methyl ester (**61**) was also evaluated against Ehrlich ascites carcinoma, meth *A, sarcoma* 180, and L1210, and was found to be more potent than 5 Fluorouracil [88].

A secoiridoid, oleuropein (**64**), mostly present in olives was reported by Hamdi et al. (2005) to have potent anticancer properties. In this study, oleuropein (**64**) was exposed to cancer cells including TF-1a, erythroleukemia; 786-O, renal cell adenocarcinoma; RPMI-7951 and RPMI-7951 melanoma cells; and LN-18 glioblastoma. Oleuropein (**64**) exhibited dose-dependent inhibition of advanced-grade tumor cell line growth and migration. Additionally, Oleuropein (**64**) irreversibly rounded cancer cells in a unique tube-disruption experiment, inhibiting their reproduction, motility, and invasiveness. In mice, 1% oleuropein (**64**) was observed to induce tumor regression in 10 mice out of 11 when administered orally between 9 and 12 days [50].

Another phytochemical study reported the isolation of known iridoids from the extracts of leaves of *Cerbera odollam* in aqueous form. Ten semi-synthetic derivatives were tested for their cytotoxicity against SKBR3 (breast), HeLa (cervical), A375 (skin), HepG2 (liver), and HCT-116 (colon). Two derivatives, namely theveside-N-(p-methoxyphenyl)-carboxamide (**65**) (IC_50_ = 190 µM) (HCT-116), and theveside-N-(cyclohexyl)-carboxamide (**66**) (IC_50_ = 150 µM) (A375 cells), displayed moderate dose-dependent cytotoxicity in HCT-116 and A375 cell lines, respectively. However, theveside 2 (**67**) (a natural iridoid) and its derivatives, exhibited no disruption of cell viability in some cell lines (cervical (HeLa), liver (HepG2), and breast (SKBR3)). The cytotoxicity of theveside 2 (**67**) was ascribed to cyclohexane as compared with the other derivatives that had aliphatic acyclic and aliphatic cyclic amides [39].

Lou et al. (2019) published a study on iridoid glycoside (Picroside II) (**68**), in which the evaluation of in vitro and in vivo anticancer properties was conducted. The outcome of the study showed that Picroside II (**68**) exhibited anti-metastatic and anti-angiogenic properties against human breast MDA-MB-231 cancer cells. When administered at 20 and 40 μM, Picroside II (**68**) displayed considerable cell migration suppression at 66.67 ± 6.32% and 44.44 ± 5.67%. Additionally, picroside II (**68**) suppressed matrix metalloproteinase 9 and downregulated a cluster of differential 31 expression in cancer cells, and angiogenesis was significantly suppressed in the chick embryochorioallantoic membrane [34].

A study on Jatamanvaltrate p (**69**) demonstrated that the compound reduced the cell viability of breast cancer cells with no toxicity in a concentration-dependent manner, inducing apoptosis and cell cycle arrest. In vivo, Jatamanvaltrate p (**69**) was administered at a dose of 15 mg/kg in mice, where a significant reduction in cancer with no toxicity was noted, with 49.7% inhibition [89].

A study on iridoids (99.5%), catalpol (**2**) (>98%), geniposide (**29**) (99%), geniposidic acid (**30**) (> 98%), and harpagoside (**70**) (99%) and their hydrolyzed forms through the action of β-glucosidase investigated their potential anticancer properties in human prostate (DU145), breast cancer cells (MDA-MB-231), and multiple myeloma (U266) was conducted. All of the five iridoids that were not hydrolyzed showed no anticancer activity while the hydrolyzed iridoids significantly showed antitumor activity. Comparatively, hydrolyzed catalpol and hydrolyzed geniposide had high potency in all cancer cell lines [90].

An in vitro study was conducted by Rathee et al. (2013) on the anticancer activities of *Picrorrhiza kurroa* extracts (PE) and their isolated iridoids, namely kutkin (**71**), picroside I (**72**), and kutkoside (**73**), against MCF-7 breast cancer cells. The research demonstrated that PE (IC_50_ = 61.86) and its isolated iridoids glycosides PS, KS, and KT had significant dose-dependent cytotoxic potential. Among the isolated iridoids, kutkin (**71**) displayed a considerable inhibitory activity of matrix metalloproteinases (MMPs) at 5 µM [91].

Pandeti et al. (2014) investigated iridoids, namely arbortristoside-A (**74**) and 7-*O*-trans-cinnamoyl 6β–hydroxyloganin (**75**), which were derived from *Nyctanthes arbortristis*, and reported considerable anticancer activities. In the same study, chemical derivation was conducted, leading to the formation of two derivatives that exhibited an improvement in anticancer activities in HepG2 (human hepatocellular carcinoma) and MCF-7 (breast adenocarcinoma) [92].

Wang et al. (2019) reported the anticancer activities of globularifolin (**76**) in SACC-83 cell lines, where the compound had an IC_50_ of 10 µM and hindered the development of the cell lines. The authors also observed that the cytotoxic effects of globularifolin (**76**) in healthy HGS cells were very minor, with an IC_50_ of 80 µM [93]. Another iridoid of interest that has been reported is gentiopicroside (**11**), which inhibits HeLa cells through multiple ways, including induced apoptosis, cycle arrest, and suppressed migration [94].

Most of the isolated and synthetic iridoids have promising anticancer properties, as reported by most in vivo and in vitro studies. Though many iridoids are reported to have cytotoxic effects, it is important to note that the safety aspects of most iridoids have not been thoroughly investigated. The published studies are mostly in vitro and in vivo, and the results obtained may not necessarily be translated into clinical settings. Some iridoids that have been isolated and synthetic iridoid derivatives reported in anticancer studies are depicted in Figure 2.

## 8. Mechanism of Action of Iridoids

Iridoids have been implicated in acting on different proteins or pathways involved in cancer progression. Two iridoids, aucubin (**1**), and geniposide (**29**), were found to cause DNA to selectively inhibit topoisomerase I without interfering with topoisomerase II [28]. Catalpol (**2**) has been cited to slow cancer progression through the down-regulation of the PI3K-Akt signaling pathway in HCT116, inhibition of TGF-β1-induced cell migration in colon cancer, and reduction of the expression of cyclin D1 in human solid tumor cell lines [95,96,97,98,99]. Additionally, valtrate (**77**) was predicted by molecular docking studies to interact with the Stat3 protein at Cys712. Additionally, valtrate (**77**) may result in a brief reduction in intracellular glutathione (GSH) levels and an increase in reactive oxygen species (ROS) [33]. A study by Wang et al. (2019) on the mechanism of action of globularifolin (**76**) revealed that the compound exhibited anticancer activities through the induction of apoptotic cell death in SACC-83 cells [93]. Another study on the iridoids aucubin (**1**), catalpol (**2**), geniposide (**29**), geniposidic acid (**28**), and harpagoside (**70**) that were hydrolyzed through the action of β-glucosidase reported considerable cytotoxicity in the hydrolyzed iridoids. It was also observed that the suppression of the proliferation of K562 cells by iridoids involved the suppression of the activation of signal transducer and activator of transcription-3 (STAT3) [90]. STAT 3 is essential in cancer progression as it regulates cell processes, such as cell survival and the cell cycle, and its suppression through hydrolyzed iridoids is pivotal in cancer management. Additionally, isolated iridoids of *Picrorrhiza kurroa*. such as Kutkin (**71)**, Picroside I (**72**), and Kutkoside (**73**) demonstrated the inhibitory activity of matrix metalloproteinases (MMPs) [91]. Few studies have explored the mechanism of action of iridoids in inhibiting cancer progression, and Figure 3 and Table 1 provide summaries of some of the iridoids that have been cited in cancer studies.

## 9. Safety Aspects of Iridoid Derivatives

The literature regarding the safety aspects of iridoids is currently sparse, as only a few studies have been published on that subject. In general, the profiles of the reported iridoids appear to be safe, and below are some of the few reported cases.

A study by Xu et al. (2015) was conducted on an iridoid-rich fraction extracted from *Valeriana jatamansi* orally administered at a dose of 3200 mg/kg body weight for an acute test. It was observed that, after 14 days, the extracts were exceedingly safe, there could be no single-dose harm, and no significant body changes were observed (*p* > 0.05). In a 90-day sub-chronic study, there were no observed adverse effects at a level of 1200 mg/kg/day. Additionally, the hematological and blood biochemical indicators and organ coefficients of the examined rats had no direct link with the toxicity of *Valeriana jatamansi* [74].

A hepatotoxicity study was carried out by Yamano et al. (1990) in rats with the iridoid geniposide (**29**) administered at 320 mg/kg mg/kg, the effects of which were compared with those of 80 mg/kg genipin (**44**). In the study, acute toxicity was also determined by evaluating the liver weight and serum levels of alanine aminotransferase (ALT), aspartate aminotransferase (AST), total bilirubin (TB), and alkaline phosphatase (ALP) levels, and it was concluded that the conversion of geniposide (**29**) to genipin (**47**) was likely to be responsible for liver toxicity [123].

A review by Zeng et al. (2020) on aucubin (**1**) highlighted that the present data (in vivo and in vitro) were not sufficient to determine the safety aspects of aucubin (**1**). Most of the reviewed studies used mice/rats, and the reported safety concerns were movement reduction and food consumption. Despite the claims, most of the reported studies have highlighted that aucubin (**1**) did not seem to cause significant toxic effects [124].

Treatment with globularifolin (**76**) also improved the expression of Bax and Caspase 3 and 9, while decreasing the expression of Bcl-2. The SACC-83 cells were also inhibited by globularifolin (**76**) at the G0/G1 phase of the cell cycle. Additionally, a cell invasion assay demonstrated that globularifolin (**76**) prevented the migration of SACC-83 cells in a concentration-dependent manner, which was similarly associated with the downregulation of MMPs 2 and 9. The JAK/STAT pathway is crucial for the growth and development of cancer cells, and this study reported that globularifolin (**76**) might suppress this pathway [93].

A study was conducted by Ma et al. (2022) on six iridoids isolated from the aerial parts of *Eucommia ulmoides*, including geniposidic acid (**28**), scyphiphin D (**78**), ulmoidoside A (**79**), and ulmoidoside B (**80**). The six compounds were investigated at a concentration range of 2.4 to 40 µM and the results demonstrated that there was no significant toxicity when the compounds were exposed to RAW 264.7 cells. The cell viability was found to be greater than 94.15% [64].

Different findings have been reported regarding the cytotoxicity of iridoids. While a majority seem to be selectively cytotoxic, other iridoids have been observed to be potentially toxic. Fukuyama et al. (2004) isolated iridoids from *Viburnum luzonicum*. The results of the cytotoxicity assay, which used the HeLa S3 (human epithelial cancer) cell line, demonstrated that two iridoid glucosides, namely luzonoside A (**81**) and luzonoside B (**82**) and their aglycons, exhibited moderate inhibitory activity, with IC_50_ values of 3–7 µM, whereas other isolated compounds showed no cytotoxicity, even at 100 µM [121]. Another in vitro study was conducted on an iridoid isolated from *Nyctanthes arbortristi*. The in vitro cytotoxic activity of Arbortristoside-C (**83**) was found to be 100 mg/mL against Hep-2 cells (human epithelial type 2) and was compared with the control culture for each concentration [125]. Another study evaluated the cytotoxicity of acetoside (**84**) isolated from *Cistanche phelypaea* on human A431 squamous carcinoma cells and Hacat keratinocytes. The collected data showed that acetonide (**84**) was not harmful to human keratinocytes at the doses studied, but that it caused minimal cell death in tumor cells (between 12 and 20%) at the concentration of 100 M [122].

## 10. Conclusions and Future Perspectives

From the reported studies, most iridoids that act as anticancer agents are from plants, and this raises concerns regarding sustainability issues and ease of regrowth. It is, therefore, crucial to consider means of obtaining plant parts sustainably without posing any significant threat to the already endangered species. However, iridoids represent a promising group of new anticancer agents; thus, more bibliometric analysis, molecular docking studies, and clinical trials are required. The current reporting format of results in studies regarding the anticancer activities of iridoids (in vitro and in vivo) varies and, though some studies use ED_50_/EC_50_, it is still difficult to fully compare the anticancer properties of iridoids that have been reported in different studies. Given that the majority of iridoids are derived from plants, choosing an isolation process is crucial as it will surely alter the yield percentage and, if not selected carefully, could pose a serious threat to species that are vulnerable to extinction. The production of synthetic compounds also requires meticulous attention, as their safety profiles still lack in many aspects. Additionally, there are still gaps in the literature regarding the mechanism of action, structure–activity relationship, in vivo adverse effects, informatics, and pharmacophore models of most iridoids. Future studies should consider the safety aspects of iridoids, especially in clinical studies, and the mechanisms of action of iridoids

## Figures and Tables

**Figure 1 cancers-15-00770-f001:**
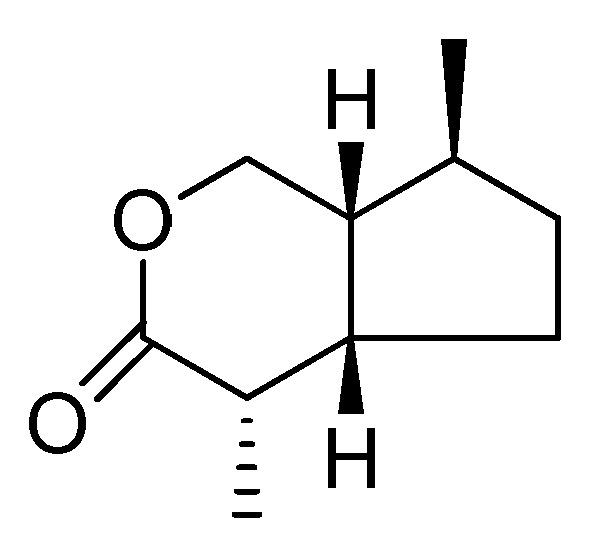
Basic structure of iridoids.

**Figure 2 cancers-15-00770-f002:**
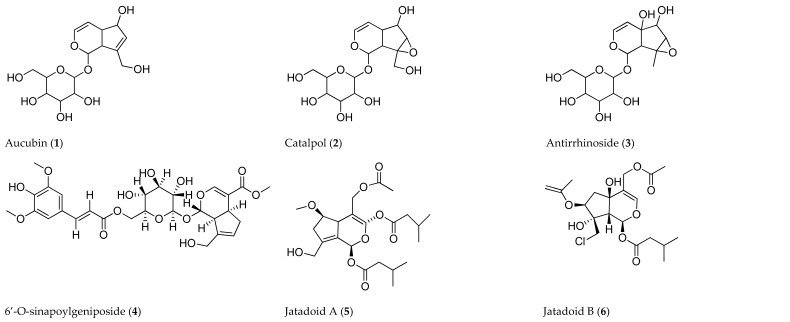
Some of the iridoids reported in isolation and anticancer studies.

**Figure 3 cancers-15-00770-f003:**
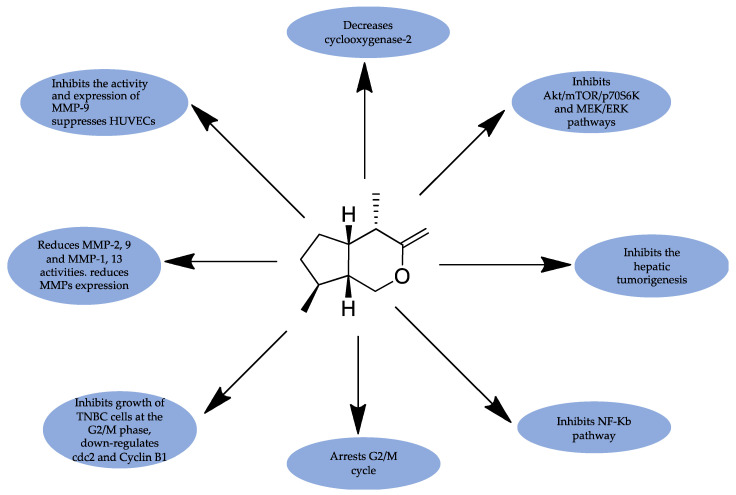
The basic structure of iridoids and their possible mechanisms of action in cancer management.

**Table 1 cancers-15-00770-t001:** Some of the iridoids that have been evaluated for anticancer properties.

Species	Iridoid(s)	Bioactivity	Target cancer	Mechanism Of Action	IC_50_ (µM)	Ref
** *Plantago asiatica* **	Aucubin (**1**)	Induces tumor cell apoptosis	Mouse 4T1 cell line and BALB/c mice	Causes DNA damage	-	[100]
** *Rehmannia glutinosa* **	Catalpol (**2**)	Inhibits cancer cell migration, induces cell apoptosis, suppresses cellular proliferation and increases apoptosis, inhibits HCT116 cell proliferation and induces cell apoptosis, inhibits cell migration, and decreases cell viability and tumor growth	OVCAR-3 ovarian cells, gastric cancer cell lines, including HGC-27, MKN-45, human HCT116 colorectal cancer cells, human NSCLC cells (A549), and MF7 cells, female BALB/c nude mice, hepatocellular carcinoma (HCC), CT26 colon cancer, and T24 human bladder cancer cells	Upregulates MicroRNA-200 and downregulates MMP-2 expression, downregulates the phosphatase and tensin homolog/PI3K-Akt signaling pathway, inhibits TGF-β1-induced cell migration and invasion of A549 cells through the inactivation of the Smad2/3 and NF-κB signaling pathways, induces and regulates protein post-translational modifications, increases miR-140-5p expression, blocks akt-mediated anti-apoptotic signaling, and inhibits expressions (IL) of IL-6, IL-8, cyclooxygenase (COX-2), and inducible nitric oxide synthase (iNOS)	-	[95,96,97,98,99]
** *Gentianella acuta* **	Gentiopicroside (**11**)	Induces apoptosis and is antiproliferative	Human ovarian carcinoma cells (SKOV3) and human cervical cancer cells (HeLa)	Causes G2/M cell cycle arrest and regulates the MAPK/Akt signaling pathway	-	[94,101]
** *Swertia chirayita* **	Swertiamarin (**14**)	Induces apoptosis	Cervical cancer cells (HeLa) and HepG2 cell line	Induces mitochondria-mediated apoptosis and targets the MEKERK signaling pathway	-	[102,103]
** *Gentiana scabra* **	Sweroside (**15**)	Induces apoptosis	Mice, leukemia P388, leukemia L12120, glioblastoma U251 cell, and PC-3 cells	Inhibits eβ-catenin transcription by suppressing TTCF/LEF activity in cells overexpressing β-catenin and downregulates the expression of Wnt downstream target genes	-	[27,104]
** *Gardenia jasminoides* **	Geniposide (**29**)	Inhibits medulloblastoma cell viability and induces cell apoptosis	Medulloblastoma cells, human breast cancer cells (MCF-7), HSC-3 cancer cells, and non-small-cell lung cancer (NSCLC) cells	Blocks the Ras/Raf/MEK/ERK pathway by downregulating miR-373 expression and regulates the PPARγ/VEGF-A pathway	-	[105,106,107,108,109]
** *Cornus officinalis* **	Morroniside (**31**)	Reverses the apoptotic effect of H2O2 on HELF cell growth, protecting cell proliferation and normal cell morphology, and inhibiting apoptosis	Human embryonic lung fibroblast (HELF) cell line and lung cancer A549 cell lines	Restores the S phase to normal levels.	-	[110]
** *Veronica* *anagallis aquatica* ** ** *Veronica persica* ** ** *Veronica thymoides* **	Catalposide (32)	Cytostatic	RD cell lines (human rhabdomyosarcoma)	Inhibits the NF-κB system	-	[82,111]
	Veratroylcatalposide (**40**)	Cytostatic	RD cell lines (human rhabdomyosarcoma)	Not specified	-	[82]
	Verproside (**33**)	Cytostatic	RD cell lines (human rhabdomyosarcoma)	Blocks the TNF-α/NF-κB signaling pathway.	-	[82,112]
	Aquaticoside C (**38**)	Cytostatic	RD cell lines (human rhabdomyosarcoma)	Not specified	-	[82]
	Verminoside (**34**)	Cytotoxic	Hep-2 cell line and human epidermoid carcinoma cells	Blocks the EMT process	128	[82,113]
	Veronicoside (**36**)	Cytotoxic	Hep-2 cell line and human epidermoid carcinoma cells	Not specified	153.3	[82]
	Amphicoside (**35**)	Cytotoxic antioxidant	Hep-2 cell line and human epidermoid carcinoma cells	Not specified	340	[82,114]
** *Valeriana dioscoridis* **	Dioscorin A (**47**), B (**50**), and C (**51**), 1-epibosnarol (**49**), 8-epi-deoxyloganin aglycone (**49**), 10-acetylpatrinoside (**52**), and 10,2′-diacetylpatrinoside (**53**)	Antiproliferative	HeLa, A2780, and T47D cell lines	Not specified	-	[83]
** *Prismatomeris tetrandra* **	Prismatomerin (**54**)	Antitumor	L929 murine connective tissue, KB-31 human cervix carcinoma, A-549 human lung carcinoma, and SW-480 human colon adenocarcinoma	Interferes with mitotic spindle formation.	-	[84]
** *Ajuga decumbens* **	8-acet-ylharpagide (**58**)	Chemopreventative	Mice	Inhibits hepatic tumorigenesis	-	[87]
** *Olea europea* **	Oleuropein (**64**)	Antimetastatic, antiproliferative, and induces apoptosis	Advanced-grade human tumors (TF-1a; 786-O, T-47D, RPMI-7951, and LoVo), mice, colorectal cancer (CRC) in C57BL/6 mice, hydroxityrosol (HT)-29 human colon adenocarcinoma cells, TPC-1 and BCPAP cells, human ER-/PR- breast cancer cells, and human MCF-7 and MDA-MB-231 cell lines	Suppresses the expression of MMP-2 and MMP-9 genes and upregulates the expression of TIMP1 and TIMP4 genes, decreases the expression levels of miR-125b, miR-16, miR-34a, p53, p21, reduces intestinal IL-6, IFN-γ, TNF-α, and IL-17A concentrations, decreases cyclooxygenase-2, downregulates hypoxia-inducible factor 1–alpha, inhibits plasminogen activator inhibitor, and suppresses the activation of the NF-κB signaling cascade	-	[31,50,115,116,117,118,119]
** *Cerbera odollam* **	Theveside 2 (**67**)	Cytotoxic	SKBR3 (breast), HeLa (cervical), A375 (skin), HepG2 (liver), and HCT-116 (colon)	Not specified	190	[39]
** *Picrorhiza kurroa* **	Picroside II (**68**)	Antiproliferative	MDA-MB-231 breast cancer cells	Inhibits the activity and expression of MMP-9 and suppresses HUVECs (tube formation)	-	[34]
** *Picrorrhiza kurroa* **	Kutkin (**71**), picroside I (**72**), and kutkoside (**73**)	Inhibits MCF-7 cell invasion and migration	MCF-7 cell lines (Human breast cancer)	Reduces MMP-2,9 and MMP-1,13 activities, and reduces MMPs’ expressions	61.9	[91]
** *Valeriana jatamansi* **	Jatamanvaltrate p (**69**)	Induces apoptosis	MDA-MB-231, MDA-MB-468, MDA-MB-453, MCF-7, and MCF-10A cancer cell lines	Inhibits growth of TNBC cells at the G2/M phase and down-regulates cdc2 and Cyclin B1	-	[89]
** *Globularia cordifolia* **	Globularifolin (**76**)	Induces apoptosis	Human glioma U87 cell line and human astrocytes, mice, salivary adenoid cystic carcinoma (SACC-83) cell line and normal human salivary gland (HSG) cell line, lung cancer A549 cancer cell line, and A549 human lung cancer cell line	Inhibits Akt/mTOR/p70S6K and MEK/ERK pathways, regulates the expression of matrix metalloproteinases (MMPs), increases Bax, Caspase 3, and 9 expression, reduces Bcl-2 expression, arrests the G2/M cycle, and inhibits the NF-Kb pathway	7.5	[26,93,120]
** *Viburnum luzonicum* **	Luzonoside A (**81**) luzonoside B (**82**)	Cytotoxic	HeLa S3 cancer cells	Not specified	-	[121]
** *Nyctanthes arbortristi* **	Arbortristoside-C (**83**)	Cytotoxic	Hep-2 cells	Not specified	-	[92]
** *Cistanche phelypaea* **	Acetoside (**84**)	Induces cell death	Human A431 squamous carcinoma cells and Hacat keratinocytes	Not specified	-	[122]

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
