# Peer review of "Iridoid Derivatives as Anticancer Agents: An Updated Review from 1970–2022"

_cancers, 2023, doi:10.3390/cancers15030770_

Round 1
Reviewer 1 Report
Major comments: The current article primarily describes iridoids and their anticancer properties. (1) Content of new insights should be further emphasized. (2) I feel “anticancer inhibitors” may be confusing. (line 2) How about “anticancer agents”, “anticancer compounds”, or any other words? (3) The authors state “many cancer have poor aqueous … in cancer patients”. (line 31-32) It sounds to me as if the bioavailability of compounds (phytochemicals) is determined only by the water solubility. Compounds that are less water-soluble but more fat-soluble and less sensitive to the first-pass effect can be expected to enter the circulating blood more efficiently. (4) Most of the experiments presented here are the in vitro studies. How about the cytotoxicity of iridoids against normal epithelial cell lines? The information should be included. (5) In case of the in vivo animal studies, how about the adverse side effects caused by iridoids? This should be addressed. (6) More in vivo studies should be included in the current article. (7) By suing iridoids, are there any human clinical trials that have been conducted so far? This should be described. (8) For readers’ convenience, it would be desirable to add the figure illustrating possible mechanism of anticancer action of iridoids. (9) Regarding the subtitle (3) (line 251), a high-throughput analysis may be included if references are available. (10) Much information on the structure activity relationship (SAR) is shown in the current manuscript. I think the informatics approach could be employed to predict an optimal or more effective structure of iridoids. Are there any studies that examine SAR by incorporating informatics? (11) Are there any studies that propose a structure based pharmacophore model? (12) The authors mention “This article, therefore, seeks to … missing gaps …”. (line 20-22) After all in this manuscript, I am not sure what is the missing gaps are. (13) In Table 1, IC50 value of each iridoid derivative should be included.
Some minor comments: (1) Typographical error: “articles” should be “article”. (2) Appropriate reference(s) should be included. (line 130) (3) TLC is generally a simple and quick method in analytical techniques. I feel the authors need to revise the sentence. (line 161) (4) Typographical error: 70 um and 103 um should be 70 uM and 103 uM. (line 265) (5) In Fig. 2, what do the numbers in parentheses stand for? The number may be somewhat confusing with in-text citations.
Author Response
Dear Reviewer,
Please see the attached cover letter!
Best regards,
Xavier

Reviewer 2 Report
The authors presented a thorough review of iridoid derivatives as potential anticancer drugs. The topics are ranging from the sources of iridoids to isolation techniques, anticancer efficacy studies, mechanism of action, and to safety considerations. This review has a good balance of breadth and depth and provides insights into the remaining challenges and future perspectives of this field. Therefore, this manuscript can be accepted.
However, careful proofreading needs to be done to solve many obvious grammar, spelling, and punctuation problems before the manuscript can be published. Some of the problems are:
1. Line 20: "This articles"
2. Line 257 and 263: "cytostostatic" (should this be cytostatic?)
3. Line 265: "texted"
4. Line 322-324 and 331-326: punctuation issues (there are more).
5. Line 517: "an that"
Author Response

(The authors gave the same response as above.)

Reviewer 3 Report
The subject of the article is natural and synthetic iridoid derivatives and their potential therapeutic roles as cancer inhibitors. To date, no bibliometric studies accessible to the authors in this regards have been published. Most studies appear to be predominantly from Asia and only a few studes were published from the United States of America, South Africa abd Brazil. However, iridoids represent a promissing group of new anticancer drugs thus more bibliometric analysis, molecular docking studies and clinical trials are required. Also, there are still missing gaps in the literature regarding the mechanism of aciton and structure-activity relationship of most iridoids.
Author Response

(The authors gave the same response as above.)

Reviewer 4 Report
The manuscript reports a very interesting overview on Irinoid compound derivatives and their use as anticabcer agents. The manuscript is well written, the topic is adequately and critically developed. In particular the Authors do report the most relevant contributions in the litterature on the subject
Author Response

(The authors gave the same response as above.)

Round 2
Reviewer 1 Report
Comment to the author(s):
1) Content of new insights should be further emphasized.
Thank you very much for the suggestion. The authors have added content on new insights. The new version (line 78 – 88) now reads:
“Iridoids have been extensively researched for their anti-tumor properties and therapeutic benefits in chronic conditions [11,21,22]. Recent reports have shown that ether terpenes can also impede DNA polymerase activity, indicating that new anti-tumor drugs derived from iridoids could be created to prevent DNA replication in cancer cells[23–25]. As extraction techniques and storage conditions have improved over the past few years, more and more iridoids have been discovered[26–29]. Review of studies on iridoids further show that these compounds may be a major focus of research in a variety of areas, including medicine and the pharmaceutical business. By enhancing antioxidant defenses and blocking the a cascades, they specifically target neurotoxicity and oxidative stress essential in disease treatment[26,30–33]. Furthermore, recent insights on iridoid research show that their key biological activity is ascribed to the inhibition of expression of multiple important proinflammatory proteins, thereby exhibiting a variety of anti-inflammatory actions [34].”
Comment: Concern addressed.
2) I feel “anticancer inhibitors” may be confusing. (line 2) How about “anticancer agents”,
“anticancer compounds”, or any other words?
The suggestion is well received and anticancer inhibitors has been changed throughout to read anticancer agents.
Comment: Concern addressed.
3) The authors state “many cancer have poor aqueous … in cancer patients”. (line 31-32) It sounds to me as if the bioavailability of compounds (phytochemicals) is determined only by the water solubility. Compounds that are less water-soluble but more fat-soluble and less sensitive to the first-pass effect can be expected to enter the circulating blood more efficiently.
The authors have looked at the suggestion and have restructured the sentence. (line 31) which now reads …
“In addition, many cancer drugs have low bioavailability thereby limiting their therapeutic effects in cancer patients.”
Comment: Regarding this sentence, references should be added and renumber in-text citations and the reference list. Additional sentences/phrases may be included in the text body.
4) Most of the experiments presented here are the in vitro studies. How about the cytotoxicity of iridoids against normal epithelial cell lines? The information should be included.
Thank you very much for the comment. Cytotoxicity studies of iridoids against epithelial cell lines have been added (580 – 593)..the section reads as follows
Different findings have been reported regarding the cytotoxicity of iridoids. While a majority seem to be selectively cytotoxic other iridoids have been observed to be potentially toxic. Fukuyama et al. (2004) isolated iridoids from Viburnum luzonicum. The results of the cytotoxicity assay which used HeLa S3 (human epithelial cancer) cell line demonstrated that two iridoid glucosides namely luzonoside A (81) and luzonoside B (82) and their aglycons exhibited moderate inhibitory activity, with IC50 values of 3-7 μM, whereas other isolated compounds showed no cytotoxicity even at 100 μM[120]. Another in vitro study on an iridoid isolated from Nyctanthes arbortristi. The in vitro cytotoxic activity of Arbortristoside-C (83) was found to be 100 mg/ml against Hep-2 cells (Human epithelial type 2) and compared withthe control culture for each concentration[124]. Another study evaluated the cytotoxicity of acetoside (84) isolated from Cistanche phelypaea on human A431 squamous carcinoma cells and Hacat keratinocytes. The collected data showed that acetonide (84) was not harmful to human keratinocytes at the doses studied, but that it did cause minimal cell death in tumor cells (between 12 and 20%) at the concentration of 100 M[121].
Comment: Concern addressed.
5)-7), 9)-11)
Comment: Concern addressed.
8) For readers’ convenience, it would be desirable to add the figure illustrating possible mechanism of anticancer action of iridoids.
We appreciate the suggestion; Figure 3 has been added.
Comment: Concern addressed.
12) The authors mention “This article, therefore, seeks to … missing gaps …”. (line 20-22) After all in this manuscript, I am not sure what is the missing gaps are.
Thank you for identifying this. The statements have been revised (line 20- 22).
“This manuscript highlights research advances that have been done to date regarding the role of iridoids and their derivatives in cancer and also underscores the missing gaps in vitro, in vivo and clinical studies .”
Comment: Concern addressed.
13) In Table 1, IC50 value of each iridoid derivative should be included.
The authors greatly appreciate the suggestion. The authors have included the IC50 values that have been reported in the table However, the reporting style of reported studies differs as not all articles use the IC50 format as a method of presenting results. The difference in the reporting style was mentioned in the conclusion as a limitation.
Comment: Concern addressed.
Author Response
Dear Reviewer,
Please see the attachment.
Best regards,
Xavier